

# Time-of-day effects in implicit racial in-group preferences are likely selection effects, not circadian rhythms

Timothy P. Schofield

Center for Research on Ageing, Health, and Wellbeing, Research School of Population Health, Australian National University, Australia

## ABSTRACT

Time-of-day effects in human psychological functioning have been known of since the 1800s. However, outside of research specifically focused on the quantification of circadian rhythms, their study has largely been neglected. Moves toward online data collection now mean that psychological investigations take place around the clock, which affords researchers the ability to easily study time-of-day effects. Recent analyses have shown, for instance, that implicit attitudes have time-of-day effects. The plausibility that these effects indicate circadian rhythms rather than selection effects is considered in the current study. There was little evidence that the time-of-day effects in implicit attitudes shifted appropriately with factors known to influence the time of circadian rhythms. Moreover, even variables that cannot logically show circadian rhythms demonstrated stronger time-of-day effects than did implicit attitudes. Taken together, these results suggest that time-of-day effects in implicit attitudes are more likely to represent processes of selection rather than circadian rhythms, but do not rule out the latter possibility.

## INTRODUCTION

Circadian rhythms refer to approximately 24-hour cycles in physical and mental processes and behaviour driven by the circadian pacemaker (*Czeisler et al., 1999*; *Panda, Hogenesch & Kay, 2002*; *Schmidt et al., 2007*). These rhythms are evidenced via robust time-of-day effects in physiology (*Jasper & Hermsdörfer, 2007*; *Refinetti & Menaker, 1992*), behaviour (*Yasseri, Sumi & Kertész, 2012*), emotion (*Golder & Macy, 2011*), and cognition (*García et al., 2012*). Although time-of-day effects have been observed in cognition since the 1800s (*Ebbinghaus, 1964*) and were shown to occur across a diverse array of tasks (*Kleitman, Cooperman & Mullin, 1933*), it was only later that they could be linked to circadian cycles (*Wever, 1979*). A demonstrable linkage of time-of-day effects to circadian cycles is of critical importance to prevent spurious conclusions about the origins of within-day variation. While processes with circadian rhythms show time-of-day effects, observational studies finding time-of-day effects are not evidence that the process has an appreciable circadian rhythm.

The consideration of circadian rhythms has largely been absent from the era of laboratory based psychological experimentation (*Schmidt et al., 2007*). Beyond basic

Corresponding author
Timothy P. Schofield,
timothy.schofield@anu.edu.au

cognitive and affective processes, circadian rhythms in psychological processes can be considered largely uncharted territory. However, with the recent explosion of online data-collection through demonstration websites (*Nosek, Banaji & Greenwald, 2002*), analyses of social media (*Golder & Macy, 2011*), and recruitment via crowd sourcing platforms (*Buhrmester, Kwang & Gosling, 2011*), behavioural scientists are now collecting data across all hours of the day. One example of where this type of data has been used to try and study circadian rhythms comes from *Zadra & Proffitt (2014)* who tested whether implicit preferences show a circadian rhythm. Implicit cognition, of which attitudes and preferences are one component, refers to thoughts which participants do not (or cannot) consciously report, but which can be measured via reaction time, priming, and choice tasks (*Fazio & Olson, 2003*).

In their work, *Zadra & Proffitt (2014)* found significant time-of-day effects in implicit preferences. These time-of-day effects followed a time-course (i.e., morning to evening) that had been anticipated for circadian rhythms in psychological processes related to cognitive control (see *Zadra & Proffitt, 2014*). The conclusion that implicit preferences have a circadian rhythm is abductively one of two reasonable interpretations of the analyses. Indeed, it is the one which both *Zadra & Proffitt (2014)* and I[1] gravitated toward. The second explanation is that in an 'always accessible' study without allocation of participants to a time of participation, that a time-of-day effect is being driven by self-selection. Self-selection may be a result of certain groups, with distinct implicit preferences, being more or less likely to participate at different times of day. In such cases, an ostensible circadian rhythm may really be a spurious side-effect of the group-level cause. Such a concern can be dealt with in different ways, such as testing to see whether the time-of-day effects differ across groups (*Zadra & Proffitt, 2014*), or by partialling out effects of groups and analysing the residuals. However, these methods both make the assumption that the groups or dimensions driving selection have been correctly identified, while also assuming that the outcome being modelled is not the factor driving selection. It may be that those with certain implicit preferences chose to participate at certain times of the day either due to an association of implicit preference patterns with chronotypes, or how they typically spend their day and the practical accessibility of the study varying across it.

Three primary endeavours were undertaken in the present paper to investigate the plausibility that time-of-day effects in implicit attitudinal preferences represent circadian processes. First, replication of the time-of-day effects in implicit attitudes was attempted using a different method to that adopted by *Zadra & Proffitt (2014)*, specifically, the variability in participant demographics was partialled out and cosinor regression performed. Next, the probability of this effect being circadian in nature was examined via analysis of moderation by factors known to influence the timing of circadian rhythms. These factors reliably include age, where older individuals show earlier circadian peaks than younger individuals (*Paine, Gander & Travier, 2006*; *Roenneberg et al., 2004*; *Van Cauter, Leproult & Kupfer, 1996*); appears to include daylight saving time (DST), where peaks occur earlier during DST (*Kantermann et al., 2007*); and sometimes includes gender, where women show earlier peaks than men (*Van Cauter, Leproult & Kupfer, 1996*). Finally, it was

[1]This was the conclusion independently drawn in the initial pre-print of this manuscript (doi: 10.7287/peerj.preprints.1475v1) prior to becoming aware of *Zadra & Proffitt's (2014)* paper on the matter.

examined whether the amount of circadian rhythmicity identified in implicit attitudes exceeded that of other variables that could only be driven by selection effects (i.e., age and gender). While not necessarily a requirement that this benchmark be exceeded, it would build confidence in the time-of-day effect being a result of more than selection.

## METHOD

### Data

Data from Project Implicit's demonstration Black-White race Implicit Association Test (IAT) was used and is described elsewhere (*Xu, Nosek & Greenwald, 2014*). This IAT uses reaction times to measure the extent to which respondents more readily mentally associate concepts of White rather than Black with positive relative to negative valence. This is an ongoing dataset, with the analyses reported here covering data collected up to 31 December 2013. The subset of adults from the US who completed the measure of implicit attitudes and indicated their race on the race codings introduced from 28 September 2006 ($N = 1,278,762$) were analysed. Frequency of participation was highest among younger adults (Fig. 1) and among women (59.66%; men: 39.87%). The observations included vary across analyses as some variables came into or out of the data-collection at different points in time. Completion time was adjusted from server time (US Eastern; F. Xu, 2015, personal communication) to local time in the respondents' county. The distribution of participation across time is presented in Fig. 2, and across space in Fig. 3. In cases where a county sat in two time-zones, it was coded as half-way between the two time-zones. In all regions observing daylight savings, the days in which a transition to or from daylight saving occurred were excluded from analysis.

Positive raw IAT scores (D) indicate a greater mental association between White and positive valence than Black with positive valence, that is, positive scores indicate a preference for White targets over Black targets. However, Americans prefer their own race over other races, and prefer White individuals over Black individuals if they are neither White nor Black (*Axt, Ebersole & Nosek, 2014*). Thus, to harmonize IAT scores across participant race, a preference for White over Black targets was interpreted as an in-group preference for all participants who were not Black, while a preference for Black over White targets was coded as an in-group preference for Black participants. This recoded score served as the dependent measure. For each categorisation of participant race, this resulted in a significant positive score. Descriptive box plots of the IAT score by race response are presented in Fig. 4. Of those who indicated their race, 71.08% indicated that they were White, and 13.47% indicated that they were Black or African American. All other groups represented less than 5% of the sample. Details on those missing a race response are noted in the caption of Fig. 4.

Preparation of the Project Implicit data was conducted in SPSS and analysis performed in STATA, with all code available at https://osf.io/um8g9/.

### Statistical models

Cosinor regression that made use of sine and cosine functions with periodicity 24 hours was employed. Cosinor regression is a highly sensitive method for the detection of

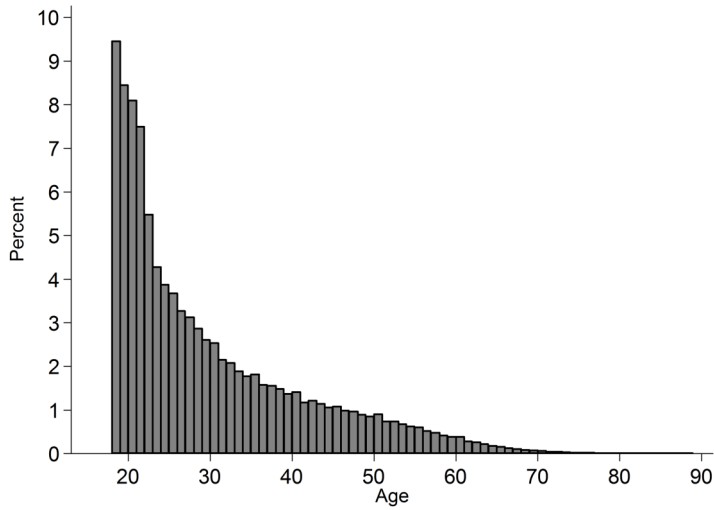

**Figure 1 Histogram of age of participants in the subset of analysed cases.** The percentage is of those who reported their age (missing: 12,000).

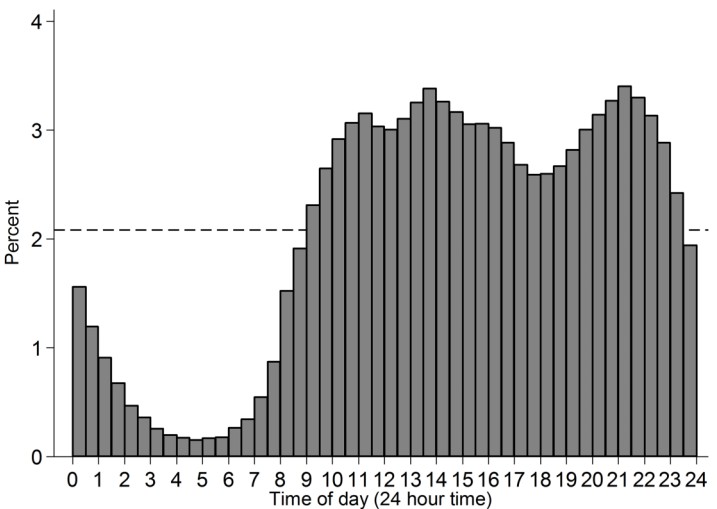

**Figure 2 Histogram of time of participation in the subset of analysed cases.** Time of day is binned into half hour intervals. Dashed reference line indicates what even distribution of participation over time-of-day would look like.

circadian rhythms, suitable for analysis of cross-sectional data even when there is a low signal to noise ratio (*Refinetti, Cornélissen & Halberg, 2007*). It should be emphasised that this type of analysis only captures a very specific wave-form and periodicity. However, even if this wave-form fits to the data, it must still make logical sense that the pattern is circadian to make such an interpretation; fit alone is not enough. No covariates were included in any of these cosinor models, instead the association of the covariate/s with the outcome were first regressed out, and the residuals subsequently analysed. These regression models included dummy codes for each level of each demographic predictor

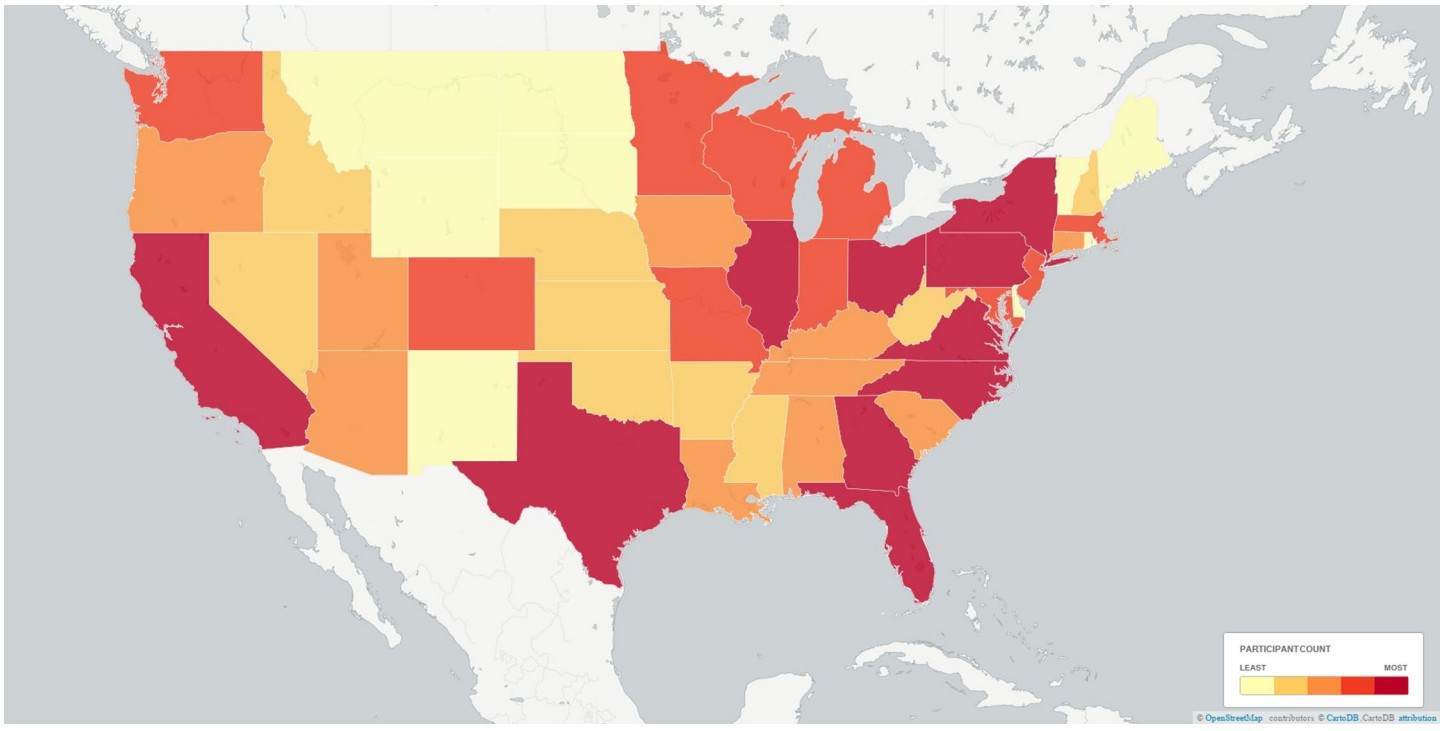

**Figure 3  Plot of participants by state binned via quantiles.** Cut-offs between bins were 5,284, 11,534, 22,077, and 39,164. Alaska and Hawaii fell into the first bin (light yellow). Note that the discrete effects of state, capturing both coarse latitudinal and longitudinal variation and socio-cultural factors, is modelled out of most analyses.

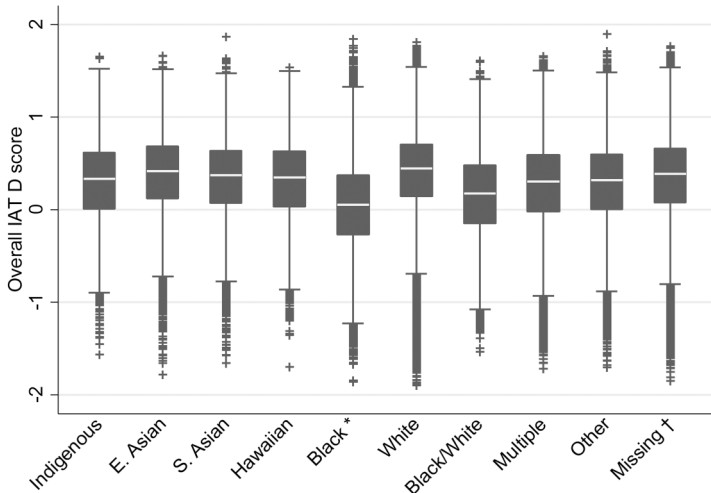

**Figure 4  Box plot of IAT D score by self-reported race of participants.** Notes: *IAT scores for Black participants are reverse scored as noted in the method. †Participants missing responses on the race variable used and required for all analyses may have responded to an older version of this variable ("ethnicity"; $n$ = 307,119) or have not responded to either ($n$ = 16,260). Those reporting being "Black–not Hispanic" on the measure used prior to 28 September 2006 were also reverse scored.

and linear and quadratic time effects to detrend the data. The educed circadian rhythm was extracted from these cosinor models using the predicted values from sine and cosine functions. Where interactions were tested the sine and cosine main effects and the main effect of the moderator were fitted along with two interaction terms; one between sine and the moderator and the other between cosine and the moderator. The joint effect of the two interaction terms were then tested.

## PART 1: REPLICATION

### Results

A significant circadian rhythm was present in the demographics adjusted model, $F(2, 1,187,494) = 470.87$, $p < .001$. This rhythm was weak, with time-of-day accounting for 0.079% of the variation in implicit preference for the in-group. Despite its weakness, the raw data and the fitted rhythm showed comparable forms (Fig. 5A). A significant circadian rhythm was also present in analyses of just the temporally detrended, but otherwise unadjusted, data on the same sample, $F(2, 1,187,494) = 301.24$, $p < .001$. This rhythm was somewhat weaker than that in the adjusted model, with time-of-day accounting for 0.051% of the variation in implicit preference for the in-group. The two rhythms followed similar time courses, with peak in-group preference at 9:10 pm in the adjusted model and at 8:27 pm in the unadjusted model (Fig. 5B).

## PART 2: CONVERGENT VALIDITY

Despite implicit in-group preferences having robust time-of-day effects, whether or not this is evidence of selection or circadian rhythms is an open question. One way to begin to answer this question is to see if the onset and timing of the time-of-day effect is affected by characteristics known to affect the onset and timing of circadian rhythms. If this rhythm was circadian in origin, it should (a) peak earlier during DST, (b) peak earlier among women than among men, and (c) peak earlier among older adults. These three hypotheses were tested in the adjusted models.

### Results

#### Daylight saving time

Moderation of the circadian rhythm by being in DST time was not observed, $F(2, 1,187,491) = 0.02$, $p = .981$, $r^2 = 0.000\%$. This null pattern replicated if analysis was limited to those regions which observe DST, $F(2, 1,164,171) = 0.10$, $p = .902$, $r^2 = 0.000\%$; or if analysis was limited just to time periods in which DST was being observed $F(2, 769,931) = 1.76$, $p = .171$, $r^2 = 0.000\%$.

#### Gender

Moderation of the circadian rhythm by gender was observed, $F(2, 1,187,491) = 22.48$, $p < .001$, $r^2 = 0.004\%$. Consistent with prior work indicating that women have earlier onset circadian rhythms than men, the rhythm observed among women had an average peak (acrophase) occurring 1 h and 18 min before that of the men (Fig. 6).

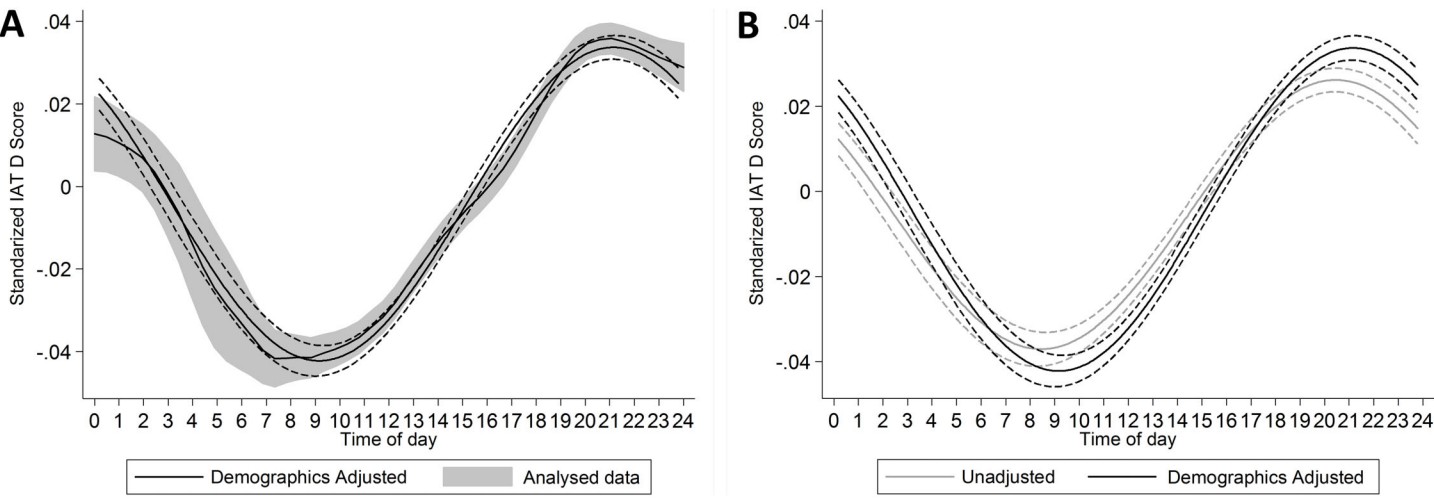

**Figure 5** (A) presents the predicted cosinor function (±95% CI) superimposed over the local polynomial fit of the raw data (±95% CI). Note, that where cosinor function takes into account time as if it was circular, the local polynomial does not, which could lead to some distortion at the ends of the function. (B) comparison of the unadjusted and adjusted cosinor functions.

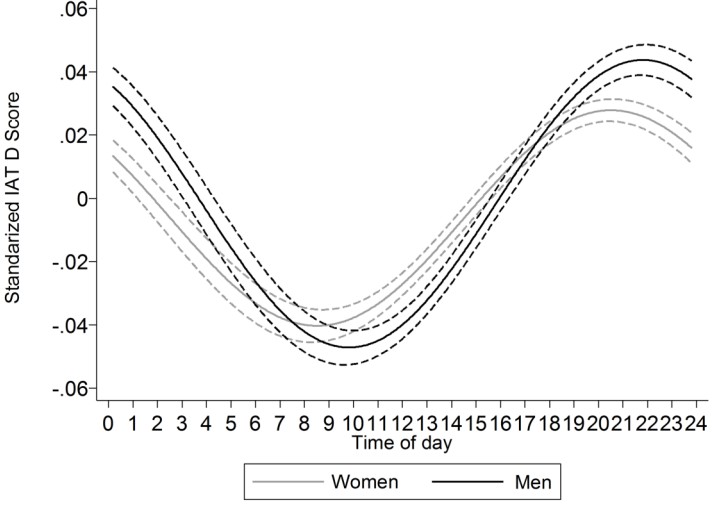

**Figure 6** Comparison of the cosinor functions (±95% CI) for women and men. The peak of the function occurs earlier in women than in men.

### Age

To avoid making assumptions about the shape of the relationship between participant age and acrophase, the circadian rhythms were first modelled for each year of age, from 18–89, separately. The predicted acrophase of each model was saved, along with the number of contributing observations. Regression of a linear age term on the predicted acrophase, weighted by the number of observations revealed a significant linear effect of age, $F(1, 1,187,495) = 64,226.46$, $p < .001$, $r^2 = 5.13\%$ (Fig. 7). For reference, about 82% of the data comes from those aged 40 or less, and given that visual inspection suggested the
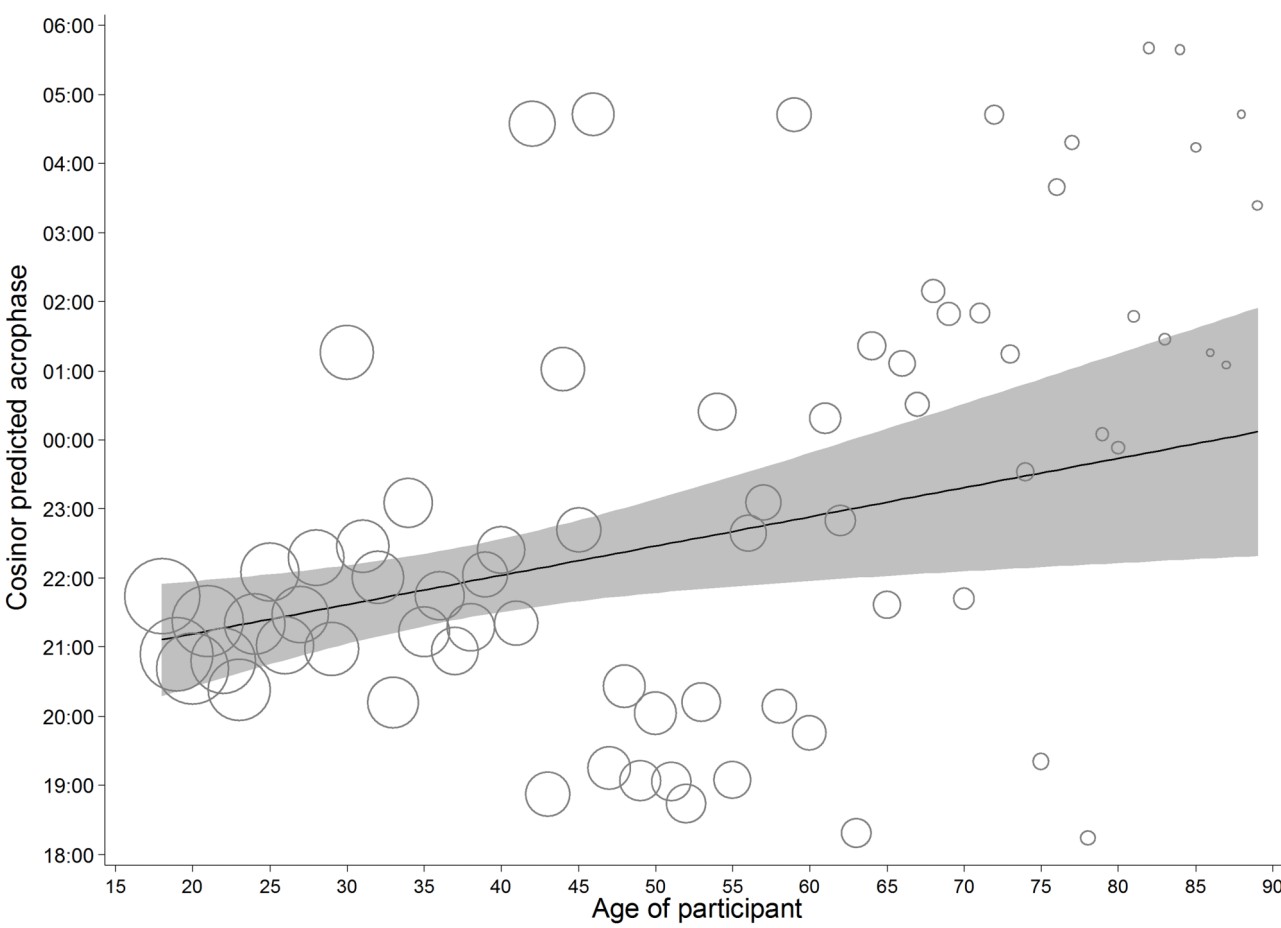

**Figure 7 Predicted acrophase of implicit attitudes for participants of each age.** Bubble area represents relative differences in the square root of the number of observations contributing to the predicted acrophase. The solid line represents the weighted regression of age on acrophase for the full sample, and the shading the 95% confidence around it.

possibility of a discrete relationship for those over 40, this analysis was replicated in those 40 or less. Again, it revealed a significant effect of similar dynamics, $F(1, 977,994) > 99,999$, $p < .001$, $r^2 = 11.66\%$. From this model, the acrophase of an individual aged 18 was predicted to occur 1 h and 7 min before that of a person aged 40. Sensitivity tests were performed, replicating the results of this model, by fitting a continuous age function into the cosinor model for those aged 18–40. The moderation by age was significant $F(2, 977,990) = 5.77$, $p = .003$, $r^2 = 0.001\%$. The predicted acrophase of an individual aged 18 was predicted to occur 1 h and 12 min before that of a person aged 40. However, in both the base tests and sensitivity analyses, this effect runs in the opposite direction to what would be expected based on the literature showing advancement of circadian rhythms with age.

## PART 3: EXCEEDING SELECTION EFFECTS

The magnitude of the cosinor effect size for implicit attitudes was compared to the cosinor effect size for two fixed demographic factors (i.e., age and gender). Neither of
these factors can vary in a wave pattern across the course of the day and so any time-of-day effect cannot be due to a circadian rhythm. Concretely, a person is not male in the morning, female in the evening, and back to male come the next morning. Similarly, chronological age does not go backwards as would be required in a circadian rhythm. Significant cosinor patterns in these fixed variables would be indicative of time-of-day selection effects rather than circadian rhythms. The magnitude of these selection effects isolated via cosinor regression was set as a benchmark that the cosinor effect in implicit attitudes needed to exceed to suggest that something more than selection was taking place.

## Results

As outlined in Part 1, the cosinor fit to implicit attitudes explained 0.079% of the variability in IAT scores. This procedure was repeated with residuals from a regression with age fitted as the outcome, and the residuals from a logistic regression with gender fitted as the outcome. The cosinor accounted for 0.547% of the variability in age (acrophase of 8:36 pm), and 0.197% of the variability in gender (acrophase of 4:37 am). The fit of the cosinor for implicit attitudes accounted for 6.92 times less variance than age despite the similar acrophase, and 2.49 times less variance than gender.

## GENERAL DISCUSSION

The finding of time-of-day effects in implicit attitudes by *Zadra & Proffitt (2014)*, was replicated here using an analysis of residuals that factored out discrete effects of socio-demographic factors. Indeed, the high conformity of the modelled cosinor to the raw data gives a degree of confidence that the time-of-day effects are robust and not artefacts of the analytic technique. The primary contribution of this research concerns the interpretation of this effect. That time-of-day effects at first glance appear like they could be circadian rhythms is not contested. But there is little evidence that the time-of-day effects are indicative of circadian rhythms. Instead, the time-of-day effects are likely better viewed as selection effects. Only one of three factors known to influence the temporal onset of circadian rhythms had the expected moderating effect, one exerted no influence, and one had an effect in the opposite direction. Moreover, even variables in the data that could only be influenced by selection showed far greater time-of-day effects than those observed for implicit attitudes. It needs to be re-emphasised here that while circadian rhythms show time-of-day effects, not all time-of-day effects are circadian rhythms. The cosinor fitting time-of-day effects in gender, for instance, may be due to differences in work hours and thus availability to participate for the average woman compared to the average man.

One possibility, which sits as a mid-way point between a true circadian rhythm and selection effects, is that the observed time-of-day effects are an artefact of participant chronotype. Chronotypes, which refer to the timing of the typical onset of sleep (*Adan et al., 2012*), would be expected to influence when participants chose to participate in the task. If chronotype was predictive of the outcome (here, in-group preferences) in some way, and chronotypes influenced time of participation then a patterning of time-of-day effects in the dependent measure could emerge in an always available study

due entirely to selection. By contrast, showing an impact of chronotypes at a fixed time of participation would suggest that there is an underlying circadian process that goes beyond selection. The most compelling arguments that the time-of-day effects in implicit attitudes are circadian in nature would come from random allocation to time of participation, or through use of constant routine and forced desynchrony procedures (*Blatter & Cajochen, 2007*).

As it stands, caution must be taken in the analysis of circadian patterns in large online data-sets which have the power to detect even very small effects. An absence of caution may be of little concern when the purpose is to document the time-of-day patterning of behaviour (e.g., *Yasseri, Sumi & Kertész, 2012*), but is likely problematic if underlying psychological and circadian processes are of interest (e.g., *Zadra & Proffitt, 2014*). Large datasets give the ability to study important effects which are statistically small (e.g., *Westgate, Riskind & Nosek, 2015*); however, caution is warranted in interpreting time-of-day effects as these may well be driven by selection rather than circadian processes.

## ACKNOWLEDGEMENTS

Thank you to A. Luckman for his comments on an early draft of the manuscript, and those who commented on the initial preprint and encouraged me to rule out selection effects.

### Funding

The author received no funding for this work.

### Competing Interests

The author declares that they have no competing interests.

### Author Contributions

- Timothy P. Schofield conceived and designed the experiments, analyzed the data, contributed reagents/materials/analysis tools, wrote the paper, prepared figures and/or tables, reviewed drafts of the paper.

### Data Deposition

The paper detailing the open data set used in this work is cited (http://doi.org/10.5334/jopd.ac), the data (https://osf.io/52qxl/), and links to all supporting code are provided in text and hosted in a third party location (https://osf.io/um8g9).

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
