# Peer review of "Time-of-day effects in implicit racial in-group preferences are likely selection effects, not circadian rhythms"

_PeerJ, doi:10.7717/peerj.1947_

## Round 0.1 · original submission · Minor Revisions

Dear Authors, Please do the necessary revisions especially the discussion section so as it can be returned for re-review,The two peer reviewers have given suggestions how to improve the manuscript.Thanking you.

Reviewer 1 ·

Basic reporting

Interesting manuscript on an important topic: are observed time-of-day effects due to underlying circadian rhythms or due to other effects (in this case a focus on selection effects). The manuscript could greatly benefit from some more elaborate explanations, mainly in the introduction section. This will make the manuscript more suited for a broader public. Specific sentences and topics which require more explanation:
Page 3 Rule 21: introduction of the term implicit preference and implicit amplitudes. Readers interested in circadian effects might not be familiar with these terms.
Page 3 Rule 23: it is unclear what is meant in this sentence: ‘a time-course that would be anticipated in circadian rhythms (i.e. morning to evening)’. Circadian rhythms refer to the oscillation in many processes caused by the core clock mechanisms. I do not see how one could anticipate a certain time-course (morning –evening) in this, other than a cosine rhythm?.
Page 4 Rule 4-5: It would be nice to have some more explanation on how self-selection can influence the observed effects. Would participants perform better at a time-point of their own choosing?
Page 4 rule 20: what are the expected effects of the three factors? How do they influence circadian rhythms? Explanation follow later on in the results section, but for understanding the approach it would be nice to read it in the introduction.

Experimental design

No comments on the experimental design. Methods are clearly explained.

Validity of the findings

Findings are clearly explained. Conclusions are drawn with the right amount of uncertainty. However, two things are missing. An elaborate description of the data demographics, including number of females/males, age distribution, country, race, and time of participation. Secondly and more optionally, a discussion on that the selection effect itself might have a circadian basis. The preference on when to do the test might be related to circadian rhythms…. In particular, since the variation observed in gender and age can for (a small part) be explained by a cosine function. Author states that a person cannot be male in the morning and female in the afternoon, still a part of the variation is explained by the cosine function, any thoughts on that?

Additional comments

-first sentence of the introduction: ‘Circadian rhythms refer to an approximately 24 hour cycle’ some word is needed after the word cycle. A cycle in what? This is explained more in the second sentence but the sentence is difficult to read this way.
- first sentence second part: circadian rhythms are not caused by homeostatic sleep pressures. Sleep is regulated via homeostatic sleep pressures and circadian rhythms. If author prefers to use both terms, than circadian rhythms should be changed to ‘time-of-day effects’. And an etc. or something like that should be included since other factors influence these time-of-day effects as well.
- page 8 rule 3: delete the word ‘only’, it is a duplication with limited.

Reviewer 2 ·

Basic reporting

The article "time-of-day effects in implicit associations are probably selection effects, not circadian rhythms" submitted to the journal, presents a very interesting topic and it really has been ignored by literature. The article is suitable for magazine.
The introduction is very clear and consistently with the subject of the article, as well as your goal. The method and results are clear, the figures are consistent with the results described.
The language is presented in a very clear, needing a little review and a more elegant format, for example the word "indeed" appears several times in the text, which may seem a poor vocabulary.
The title does not clearly show the goal and the result of the article. I suggest the removal of the word "probably" not conclusive to appear in a title.

Experimental design

The method applied is consistent and the statistic is correct.

Validity of the findings

In my opinion, the discussion presents the biggest problem because it is poor. And does not explore the impact of the results found, its importance and what is the selection effects and the difference between circadian rhythms. The discussion should be more substantial. The author shows a result that distinguishes between two phenomena, and points one to cause a psychological effect, so the discussion should provide subsidies to completion.

---

## Round 0.2 · Minor Revisions

Dear Authors, Good news.

There are very minor comments. Please revise the statements in the introduction section that the peer reviewer requests be edited and resend so that the manuscript can be accepted.

Reviewer 1 ·

Basic reporting

Author has performed a nice revision of the manuscript and all major concerns were addressed. In particular, the readability for a broader public is greatly improved. This paper is now suited for publication.
One final minor comment:
The sentence in the introduction section starting with "These time-of-day effects…" (r 11) would benefit from some reference(s) where this statement has been postulated before. The fragment in this sentence: 'that had been anticipated' suggests it coming from a previous source.

Experimental design

no comments

Validity of the findings

no comments

Additional comments

no further comments

---

## Round 0.3 · accepted · Accept

Dear Author, Thank you for the minor revisions that have now allowed us to accept your manuscript. Congratulations.